# When Answers Change: Testing Sensitivity of Language Models as Decision Makers

## Abstract

Large language models are typically evaluated on static benchmarks before deployment. Based on their accuracy across a variety of tasks, models are often claimed to exhibit incremental improvements in intelligence. However, this evaluation paradigm poses deployment risks given the growing use of language models in critical domains such as medicine, law, and scientific analysis. In this study, we identify gaps in current evaluation practices and conduct a systematic analysis across four models and seven domains using three types of perturbations: (1) invisible, (2) syntactic, and (3) occlusion; they vary the presentation of information to the model. Invisible perturbations introduce changes imperceptible to humans, syntactic perturbations modify the surface form while preserving semantics (meaning), and occlusion perturbations alter the available information by masking important tokens.

Reliable assistive decision-making involves two key properties: decision invariance and confidence calibration. Decision invariance requires that model predictions remain stable when same questions are asked to the model with minor presentation changes. Confidence calibration measures whether a model appropriately expresses uncertainty when it unsure about its predictions.

Across seven domains, the four models displayed substantial instability in our experiments. The invisible perturbation and syntactic perturbation for which the models should have been invariant had a flip rate of 2.9% to 18.3%. However, the model's accuracy does not reflect this instability, since the correct-to-incorrect and incorrect-to-correct transitions are nearly equal. With occlusion perturbation, which hides key information from the model, the model remained highly confident (between 80%-90%) when it changed its answer. These results demonstrate that benchmark correctness alone does not capture the reliability of language models and suggest that stability and calibrated uncertainty should be treated as the primary evaluation criteria for deployment in decision-support settings.

## 1 Introduction

In recent times, language models (LM) have been deployed across various domains to make decision recommendations to people. The domains include, but are not limited to, medical decision support (Singhal et al., 2025; Nori et al., 2024), legal assistance systems (Bommarito II & Katz, 2022), education or certification systems (Kung et al., 2023; Kasneci et al., 2023), and public service chatbots (Ferrara, 2023b; Talukder, 2025). The overall premise sounds promising. However, the capabilities of modern AI systems are not fully captured by benchmark evaluations. With every new iteration of models, some benchmarks become obsolete, and a new benchmark emerges, designed to make the same tasks harder. Some popular examples include GLUE (Wang et al., 2019) to SuperGLUE (Wang et al., 2020), SQuAD (Rajpurkar et al., 2016) to SQuAD 2.0 (Rajpurkar et al., 2018), MMLU (Hendrycks et al., 2021) to MMLU Pro (Wang et al., 2024), and so on. To some, this may indicate progress in AI and an improvement in intelligence, but it may also mean that these benchmarks overlook important aspects of model behavior. Also, the primary metric for most benchmarks is accuracy, F1-score, pass@k, or exact match.

As mentioned before, these LMs are now used to support decision-making at a large scale across several sensitive domains; therefore, we need to ensure they adhere to the concept of *Decision Invariance* (Von Neumann & Morgenstern, 1947). This principle states that the decision should remain stable if the underlying information remains unchanged, even if the input representation changes. In the context of LMs, this means that any linguistic differences should not affect the model output as long as the information remains the same. Regarding AI policy, there are four key aspects: fairness, consistency, predictability, and auditability (Diakopoulos, 2016; Doshi-Velez & Kim, 2017; Smuha, 2019). Decision invariance relates to fairness, consistency, and predictability, three of the four aspects. Therefore, ensuring decision invariance is central to building trustworthy AI systems that operate reliably in real-world settings.

At present, most benchmarks consist of questions, and the model is prompted to generate the answers. However, in the real world, it is a more nuanced process because humans use different grammatical structures based on many latent factors. The framing of questions depends on the user's educational, linguistic, and cultural backgrounds. From a benchmark perspective, these behavioral variations often go unnoticed. The model may produce a correct answer if the question is formatted a certain way because the linguistic pattern exists in the pre-training data; however, even slight linguistic changes can cause it to fail (Sclar et al., 2024). If such variations lead to different model decisions, then these systems violate the principle of decision invariance and raise concerns for deployment in decision-support settings.

One way to evaluate decision invariance is to test whether model predictions remain stable under semantically preserving perturbations of the input. In addition, model robustness or calibration can be examined under information masking that removes important cues from the input. In this paper, we study the answer instability of language models under controlled perturbations. We analyze three different axes of perturbing information:

- Invisible perturbation: Adding non-printing Unicode characters in the text.

- Syntactic perturbation: Changing the syntactic arrangement without changing the semantics.

- Occlusion perturbation: Masking important tokens from the question to evaluate model sensitivity to missing information.

The first two are invariance-type perturbations, meaning, in terms of information, nothing changes, and the third perturbation is actually withholding information, which should lead to a decline in accuracy and reduced confidence in the model's new answer.

Across 4 models and 7 datasets, we observe a clear gap between benchmark performance and answer stability under perturbation. Invariance-type perturbations, such as invisible characters and grammatical rewrites, reveal sensitivity to superficial input variation. Occlusion perturbations produce the largest instability; removing a small number of important information-bearing tokens may change the model's answer, but if the model is well calibrated, then its confidence should also go down, accompanied by the failures. This gap between benchmark accuracy and prediction stability suggests that correctness alone is an insufficient indicator of reliability for language models deployed in decision-support settings.

The contribution of this paper is an empirical analysis of pathologies in recent language models rather than algorithmic. We characterize a previously under-measured reliability failure. We provide a compact, robust characterization of behavior in terms of model information sensitivity across domains, and we argue that stability metrics should be treated as first-class evaluation targets rather than secondary diagnostics.

## 2 Background

Language models are increasingly used in assistive decision-making settings, where users may rely on model outputs to support judgments in domains such as healthcare, law, education, and science (Ferrara, 2023a; Tamkin et al., 2023). In such settings, reliability is not only a matter of aggregate accuracy, but also of whether model behavior remains consistent under equivalent ways of presenting the same information. Prior work on algorithmic decision support has shown that people often over-rely on automated systems, making

inappropriate trust calibration a practical concern (Bansal et al., 2021; Dietvorst et al., 2015; Lee & See, 2004). For language models, this concern is amplified by the opacity of their internal decision process and the limited reliability of their self-reported confidence (Turpin et al., 2023; Kadavath et al., 2022; Zhang et al., 2024; Wang & Stengel-Eskin, 2025).

A related line of work studies behavioral robustness under controlled perturbations. Behavioral testing frameworks such as CHECKLIST evaluate whether model predictions remain invariant under meaning-preserving changes to the input (Ribeiro et al., 2020). Challenge datasets such as HANS and PAWS similarly expose heuristic shortcuts and sensitivity to superficial variation (McCoy et al., 2019; Zhang et al., 2019). More recent work extends robustness analysis to modern language models and prompt sensitivity, showing that model outputs can vary substantially under small input reformulations (Goel et al., 2021; Sclar et al., 2024; Wang et al., 2024). However, less attention has been given to prediction stability in decision-support settings, where semantically equivalent formulations should ideally yield the same decision, and where confidence should decrease when critical information is removed.

Our study builds on this behavioral testing perspective. We evaluate whether language models satisfy decision invariance under semantic-preserving perturbations and whether their self-reported confidence responds appropriately when information-bearing content is occluded. We focus on multiple-choice question answering in critical domains spanning healthcare, law, ethics, economics, and science.

## 3 Methods

In this section, we provide a detailed description of our workflow and the intuitions behind perturbations, model and dataset choices, and evaluation metrics.

**Data**  To quantitatively ground our assumptions, we select a subset of benchmark datasets representative of critical domains. Overall, our evaluation set contains a total of 4098 multiple-choice questions taken from MedQA (Jin et al., 2020) and the MMLU subsets of law, ethics, economics, biology, chemistry, and physics (Hendrycks et al., 2021).

**Models**  To show that the observations are agnostic to LMs, we use four different LMs of various sizes. The four models are: Llama-3.1-8B (Grattafiori et al., 2024), Gemma-2-9B (Team et al., 2024), Llama-Scout-17B (Adcock et al., 2026), and Moonshot Kimi-K2-1T (Team et al., 2025). All of these models are open-weight models, and we are using the instruction-tuned variants. The first two models were hosted locally, and the latter two were accessed via Groq API. To correctly analyze sensitivity, all inferences from these models use greedy decoding (temperature set to 0) to maximize consistency. We only evaluate the zero-shot scenario because most end users use these models for ad hoc decision-making.

| **Original Question** |
|---|
| The quantum efficiency of a photon detector is 0.1. If 100 photons are sent into the detector, one after the other, the detector will detect photons |
| **Invisible Perturbation** |
| The quantum efficiency of a photon detector is 0.1. If 100 photons are sent into the detector, one after the other, the detector will<U+2060> detect photons |
| **Syntactic Perturbation** |
| If 100 photons are sent into the detector one after the other, and the quantum efficiency of the detector is 0.1, how many photons will it detect? |
| **Occlusion Perturbation** |
| The quantum efficiency of a *blank* is 0.1. If *blank* are sent into the detector, one after the other, the detector will *blank* |

Table 1: Example perturbations applied to a physics question taken from the actual perturbed dataset.

> You are rewriting multiple-choice question stems to change their **grammatical structure** while keeping the **meaning identical**.
>
> **Original question stem:**
> "{stem}"
>
> **Your task**
>
> Produce **{n} alternative stems** that satisfy the following requirements:
>
> • Use a **different grammatical structure** (e.g., active vs. passive voice, clause reordering, subject–object reordering, etc.).
> • **Preserve the exact meaning and truth conditions** of the original question.
> • **Do not change** any technical terms, named entities, or numerical values.
> • **Do not add or remove** any facts.
>
> **Constraint**
>
> Assuming the **answer options remain exactly the same**, and the **correct option must remain correct** for all rewritten stems.

Figure 1: Prompt to Gemini-2.5 for perturbing the original stem question syntactically without causing any loss of information.

**Perturbations**   To analyze the sensitivity of LMs relative to information quality and presentation. To do this, we use three types of perturbation that vary the degree of information obfuscation (see Table 1). The details of these perturbations are:

- **Invisible perturbation:** The information presentation and quality are both preserved in this type of perturbation. One non-printing Unicode character is inserted into the question at random word boundaries. Three instances of such perturbed questions are created for each original question. This targets the tokenizer behavior and the model behavior, as they are coupled.

- **Syntactic perturbation:** We change the surface presentation of the question grammatically without changing the actual information. To do this, we ask Gemini-2.5-Flash to change the question stem grammatically while preserving the truth condition of the original stem. The exact prompt can be found in Fig 1. We generated three instances of these perturbations for each question. A concern would be that these rewrites would alter the information in the questions too much; therefore, we use an embedding model (miniLM-L6-v2) to measure semantic similarity between the original and perturbed questions, and filtered out those with a cosine similarity below 0.9. This filtering removes 19.69% of grammar rewrites. This is a heuristic step that reduces semantic drift, but this is more of a pragmatic step that filters out bad rewrites rather than establishing semantic equivalence. To verify that the remaining perturbations meet standards, we randomly sample 100 perturbed questions along with their original stems and manually annotate their semantic equivalence. We used a three-class annotation schema: (1) "good": meaning was preserved, only superficial changes to syntactic structure; (2) "bad": the meaning of the question changed; and (3) "undetermined": meaning changed slightly, but it is unclear whether it was relevant to answer the question. We found that 90 of 100 perturbations were good, 5 were bad, and 5 were undetermined. We observe that most of the questions underwent rewording, or change in voice, or change in sentence order. We could not find any perturbation with hallucinated facts. The intuition behind this perturbation is that people present/structure information differently based on their socio-linguistic background.

- **Occlusion perturbation:** A RAKE (Rose et al., 2010) style keyword extraction was used to identify information-bearing words and phrases. The top-5 words or phrases are used as candidate informational tokens, which are randomly masked from the question stem to create three perturbations per question. As a fail-safe for RAKE when it produces very few words and phrases, we use parts-of-speech tags such as noun, proper noun, adjective, verb, and number as candidate tokens to mask. The intuition here is to analyze how much information-bearing words actually matter to LMs and whether, if they make mistakes, they do so with lower confidence than when they answer correctly.

**Evaluation Metrics**

- **Flip rate:** This metric measures the normalized proportion of answers that change when the question stem is perturbed. It provides an estimate of the model's sensitivity to a particular perturbation type. For perturbations that preserve information (invisible and syntactic), a stable model should exhibit a low flip rate. For occlusion perturbations, where information is intentionally removed, higher flip rates are expected.

- **Correct-to-incorrect rate ($C->I$):** When the model prediction changes under perturbation, the outcome can transition between several states (e.g., incorrect-to-correct, incorrect-to-incorrect, or correct-to-incorrect). This metric measures only the proportion of cases where a model's originally correct answer becomes incorrect after perturbation. This captures the most critical failure mode for decision-support systems. For information-preserving perturbations, this rate should remain low.

- **Mean confidence change on flip ($\Delta\,Conf(Flip)$):** The model is instructed to provide both an answer and a confidence estimate for that answer. For cases where the predicted answer changes under perturbation, we measure the average change in the reported confidence between the original confidence and the perturbed confidence. This metric assesses whether the model's confidence accurately reflects the instability of its predictions.

- **Mean confidence change from correct to incorrect:** This metric focuses specifically on cases where the model transitions from a correct answer to an incorrect answer after perturbation. It measures the average change in reported confidence across these instances. Large negative confidence would indicate better alignment between model confidence and prediction reliability.

## 4    Result & Analysis

The result for the models for various perturbations can be seen in Table 2. We analyze the result based on the type of perturbation and its effect on the LMs.

| Model | Perturb. | Orig. Acc | Pert. Acc | Flip | C→I | ΔConf (Flip) | ΔConf (C→I) |
|---|---|---|---|---|---|---|---|
| Llama3.1-8B | invisible | 0.478 | 0.478 | 0.029 | 0.009 | -0.014 | -0.008 |
| | syntactic | 0.478 | 0.470 | 0.183 | 0.064 | 0.000 | -0.007 |
| | occlusion | 0.476 | 0.424 | 0.293 | 0.117 | -0.020 | -0.029 |
| Gemma2-9B | invisible | 0.498 | 0.499 | 0.023 | 0.008 | -0.001 | -0.002 |
| | syntactic | 0.498 | 0.500 | 0.169 | 0.065 | -0.003 | -0.004 |
| | occlusion | 0.502 | 0.446 | 0.221 | 0.104 | -0.009 | -0.011 |
| Llama-Scout | invisible | 0.694 | 0.694 | 0.040 | 0.016 | -0.001 | -0.001 |
| | syntactic | 0.694 | 0.701 | 0.121 | 0.052 | -0.001 | -0.004 |
| | occlusion | 0.694 | 0.628 | 0.197 | 0.113 | -0.015 | -0.019 |
| Kimi-K2 | invisible | 0.714 | 0.715 | 0.157 | 0.057 | -0.003 | -0.013 |
| | **syntactic** | 0.714 | **0.716** | **0.182** | 0.073 | -0.009 | -0.017 |
| | occlusion | 0.714 | 0.642 | 0.251 | 0.133 | -0.062 | -0.083 |

Table 2: Prediction stability of four language models under invisible, syntactic, and occlusion perturbations. Flip denotes the fraction of predictions that change under perturbation, while C→I measures cases where originally correct predictions become incorrect.

**Invisible Perturbation:**    The results table shows that three of the four LMs are quite robust to this type of perturbation with a flip rate of 2-4%. However, the largest LM Kimi-K2 showed sensitivity to this type of perturbation, with a flip rate of about 16%. While we do not have enough data to claim that model size is directly related to sensitivity to this type of perturbation, we can see that the two larger models show the highest flip rates. These perturbations preserve both the semantic meaning and surface appearance of the input, indicating that even small tokenization-level variations can influence model predictions.

Even when predictions change infrequently, a portion of these flips convert originally correct answers into incorrect ones. Under invisible perturbations, the C→I rate ranges from 0.8% for Gemma2-9B to 1.6% for Llama-Scout, while Llama3.1-8B shows a similar rate of 0.9%. The Kimi-K2 model again stands out with a substantially higher C→I rate of 5.7%. Similar to the flip rate, the two larger models show a higher number of correct-to-incorrect transitions.

Self-reported confidence of these opaque models is poor, as evidenced by $\Delta Conf(Flip)$ and $\Delta Conf(C->I)$. While we can see that confidence decreased for perturbed flipped answers (denoted by the negative sign), the average change in confidence remained very small. Gemma2 and Llama-Scout confidence changes by only 0.1%. Llama3.1 shows a larger decrease in confidence of about 1.4%. Even for Kimi-K2, which experiences the largest instability, the confidence change on flip is only 0.3%. A similar pattern can be seen for $\Delta Conf(C->I)$, and the changes are quite nominal for all models. Interestingly, Kimi-K2 showed a larger change in confidence, rising to 1.3%. Llama-Scout and Gemma2 remained almost the same, with little change in mean confidence. The confidence change decreased for Llama3.1 from 1.4% to 0.8%. This suggests that self-reported confidence does not reliably track failure under invisible perturbations.

**Syntactic Perturbation:** Despite the perturbation's lossless nature, the LMs struggled to remain stable. Flip rates range from 12.1% for Llama-Scout (the lowest) to 18.2% for Kimi-K2 (the highest). In spite of the high flip rates, the difference between original and perturbed accuracy is quite nominal, suggesting there are several incorrect transitions to correct, too. This result suggests that people from different backgrounds will get different results depending on how they grammatically formulate the question.

Further instability is highlighted by the C->I rate across models, which ranges from 5.2% to 7.3%, and by the model behavior remaining proportional to the flip rates. The results suggest that linguistic variation in conveying the same information may lead LMs to transition from correct to incorrect decisions. The sensitivity is also much higher for syntactic change. In decision-support contexts, such reversals undermine the principle of decision invariance and introduce instability into otherwise correct predictions.

Confidence behavior under syntactic perturbations suggests that models do not meaningfully signal this instability. The mean confidence change when predictions flip remains very small across models, ranging from approximately 0.9% to 0%. Similarly, the confidence change for correct-to-incorrect transitions is modest, ranging from 0.4% to 1.7%. These small shifts indicate that models maintain nearly the same confidence levels even when syntactic variation causes their answers to change or become incorrect. As a result, confidence estimates provide limited warning that predictions may be sensitive to grammatical reformulation.

**Occlusion Perturbation:** As expected, accuracy decreases, and flip rates increase across all models. The accuracy drop ranges from approximately 5% to 7%, while flip rates increase to 19.7%-29.3%.

Correct-to-incorrect transitions also increase relative to syntactic perturbations. For example, the highest C→I rate is observed for Kimi-K2 at 13.3%, compared to 7.3% under syntactic perturbations. Across models, the increase in C→I transitions is approximately five to six percentage points. Although occlusion introduces additional instability, the magnitude of this increase is relatively modest compared to the instability already observed under syntactic perturbations. This suggests that models are already highly sensitive to surface-level variation.

Confidence changes are larger under occlusion perturbations than under the other perturbation types. The largest decrease occurs for Kimi-K2, where confidence drops by 6.2% when predictions flip and by 8.3% when correct answers become incorrect. These results indicate that models partially adjust their confidence when information is removed, but the magnitude of this adjustment remains small relative to the level of prediction instability. Consequently, confidence estimates provide only a limited indication that model predictions have become unreliable when key information is missing. These findings suggest that language models rely heavily on explicit lexical cues when forming predictions, but their confidence signals remain weak indicators of when critical information is missing.

## 5    Discussion

Our results demonstrate a few key-points, summarized below.

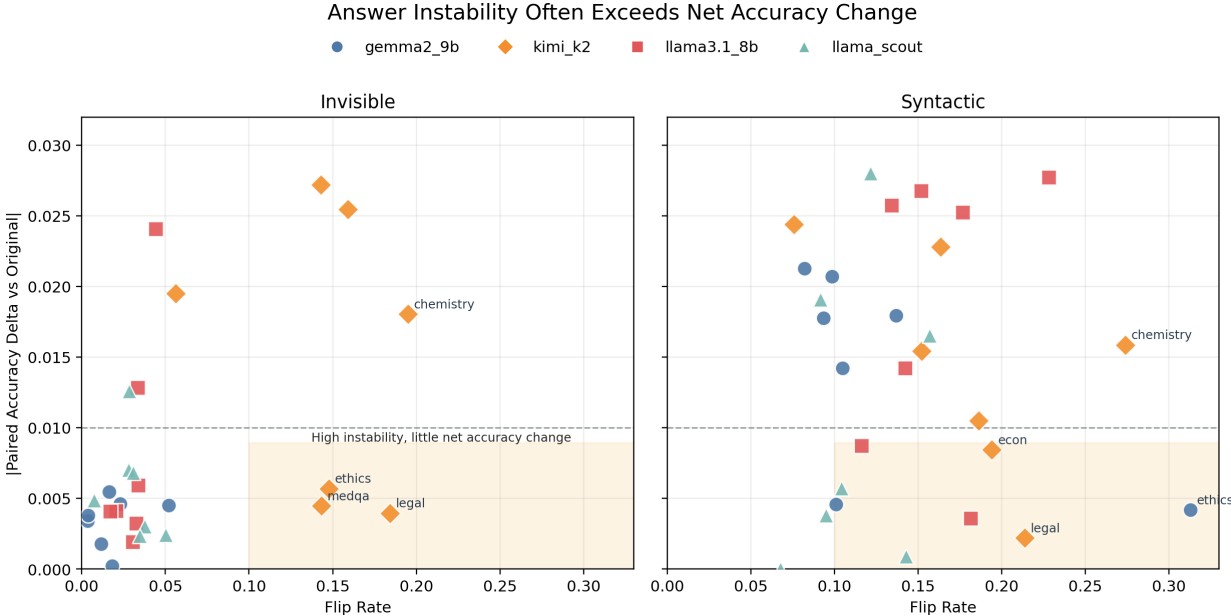

Figure 2: Relationship between answer flip rate and paired accuracy change across models and datasets. The shaded region highlights cases where perturbations cause substantial instability in the answer while producing little net change in accuracy.

**Accuracy Masks Instability**    The results reveal high flip rates for some models, while their accuracy showed nominal change (as in Figure 2). The shaded region in the plot denotes the space where models showed no net change in accuracy but had high flip rates on perturbations that should have been decision-invariant. If a benchmark relies solely on a model's accuracy, it will fail to account for how unstable the model can be under slight changes in the way information is presented.

**Linguistic Variation Can Lead to Non-Uniform Prediction Instability Across Domains**    We observe relatively high flip rates for syntactic perturbations (Table 2), even though these perturbations are designed to preserve the semantic meaning of the question. On average, the flip rate across models is approximately 16%, indicating that a substantial fraction of predictions change when the surface form of the input is modified. Such sensitivity to linguistic variation raises concerns for real-world deployment scenarios, where users may phrase the same question in multiple ways.

To further examine domain-level behavior, we visualize the correct-to-incorrect transition rates across models and domains in Figure 3. The results show that prediction instability is not uniform across tasks. Domains such as law and ethics exhibit higher correct-to-incorrect transition rates, whereas biology appears comparatively more stable. These findings suggest that tasks requiring interpretive reasoning or contextual judgment may be more sensitive to linguistic variation than domains where questions are more structurally constrained. These results highlight that prediction stability can vary substantially across application domains, suggesting that robustness evaluations should consider task-specific behavior rather than assuming uniform model reliability.

**Self-Reported Confidence in Language Models Is Unreliable**    Figure 4 examines model confidence under occlusion perturbations by comparing two types of prediction transitions: correct-to-incorrect (C→I) risky flips and incorrect-to-correct (I→C) recovery flips. These recovery transitions suggest that models may

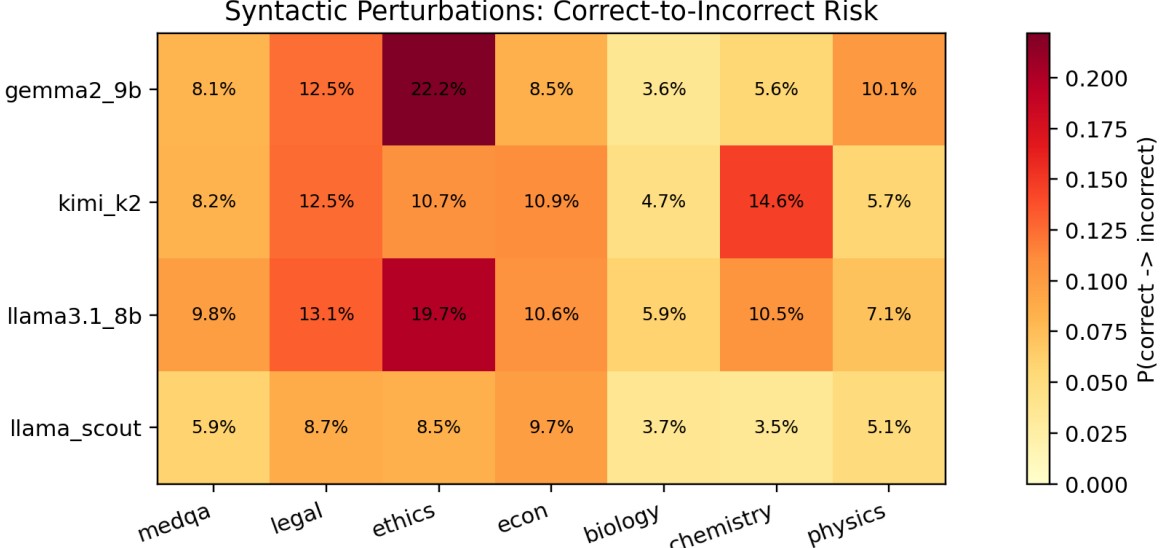

Figure 3: Correct-to-incorrect transition rates across domains under syntactic perturbations.

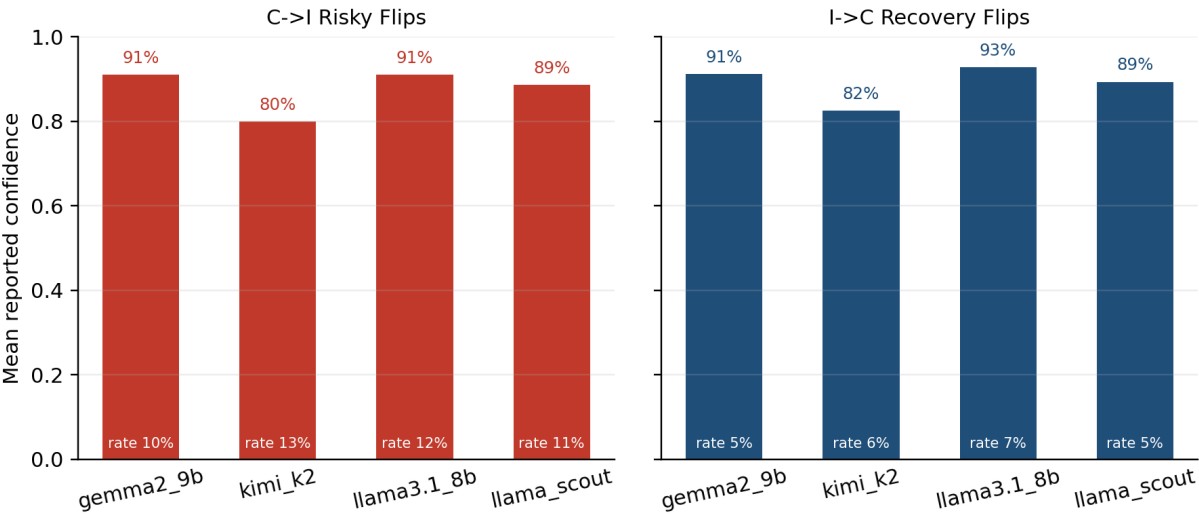

Figure 4: Model confidence under occlusion perturbations. The left panel shows mean confidence for correct-to-incorrect (C→I) risky flips, and the right panel shows incorrect-to-correct (I→C) recovery flips. Confidence remains high in both cases, indicating that language model confidence does not reliably signal when predictions become unreliable after key information is removed.

rely on contextual cues or partial information even when information bearing words are masked. Confidence levels of recovery flips range between 82% and 93%. This is slightly higher than model confidence on risky flips which range between 80% to 91%. A well-calibrated model should exhibit two behaviors: (1) dip in confidence that is proportional to the flip-rate; (2) risky flips should be accompanied by low confidence. The first behavior is missing and we have established that in Table 2. The second behavior is also missing because risky flips and recovery flips have nominal difference of confidence estimate.

This suggests that self-reported confidence estimates in the tested language models are poorly calibrated even when information is degraded. In decision-support settings, such miscalibration can be problematic because users may interpret high confidence as a signal of reliability. Our results show that even when key information is removed, and predictions become incorrect, models often maintain nearly the same confidence levels.

## 6 Conclusion

The motivation for this work is to better understand language model pathologies in critical-domains when questions are presented differently. While benchmark accuracy measures aggregate correctness, decision-support systems also require prediction stability under equivalent inputs. Across four models, we find that overall accuracy often changes only marginally under perturbation, even when answers flip rapidly, and a substantial amount of those flips are from correct-to-incorrect transitions. This gap shows that benchmark accuracy alone does not capture an important dimension of model reliability and reveals violations of decision invariance.

Our results further show that surface-level syntactic variation can cause models to change their decisions even when the underlying meaning of the question is preserved. This poses a practical deployment risk, since users naturally differ in how they phrase questions based on linguistic, educational, and cultural backgrounds. In such settings, semantically equivalent requests may receive different model outputs, undermining consistency, predictability, and fairness in decision-support applications.

Under occlusion perturbations, we expected models not only to make more errors when important information was removed, but also to reflect this degradation through lower self-reported confidence. Although confidence drops were somewhat larger under occlusion than under invariance-type perturbations, the mean confidence change is nominal relative to the observed instability. This suggests that self-reported confidence estimates are not yet reliable enough to serve as actionable indicators of prediction failure.

Overall, our findings argue that stability should be treated as a first-class evaluation target for language models. Metrics such as answer flip rate, correct-to-incorrect transition rate, and confidence sensitivity under perturbation should complement correctness in pre-deployment evaluation, especially in critical domains where consistent and dependable model behavior matters as much as raw correctness.

## 7 Limitation

This study is an empirical analysis of language model behavior under controlled perturbations and builds on previously proposed behavioral testing frameworks. The contribution of this work lies in the systematic evaluation of prediction stability across multiple domains rather than the introduction of a new methodological framework. The experiments were conducted on four representative language models spanning different sizes and architectures. While this selection provides a useful cross-section of modern models, evaluating a larger, more diverse set would allow stronger generalization of the observed patterns. Future work could extend this analysis to additional model families and larger evaluation suites to better understand the prevalence of prediction instability across modern language models.

Our analysis focuses on multiple-choice QA tasks in order to probe the stability of model behavior under input variation. Although such tasks are common in benchmark evaluations, real-world decision-support systems may involve more complex reasoning processes or open-ended responses. The perturbations considered in this study represent a limited subset of possible input variations. Other forms of variation, such as human-written paraphrases, adversarial prompt modifications, or cross-lingual inputs, may reveal further instability.

Finally, we did a manual error analysis on a small subset of syntactic perturbation after a heuristic filtering. We found about 10% error-rate and that has potentially propagated to our results.

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
