# OpenReview forum: "When Answers Change: Testing Sensitivity of Language Models as Decision Makers"
_TMLR — Under review for TMLR_

### Review · Reviewer_1otq · 2026-04-22

**Summary Of Contributions:**

This paper studies answer stability in multiple-choice question answering under three input perturbations: invisible Unicode insertion, syntactic rewrites, and token occlusion. The experiments cover four open-weight language models and seven benchmark domains drawn from MedQA and MMLU. The central empirical finding is that paired answer flips can be substantial even when net accuracy changes little, so benchmark accuracy alone can hide instability. The paper also reports that self-reported confidence remains high on many occlusion-induced failures.

The strongest aspect of the paper is the clarity of the core empirical message. The main weaknesses are novelty positioning, underspecified evaluation methodology, and claims that extend beyond what the experiments actually support.

**Audience:**

Yes

**Audience Explanation:**

Stability under semantically equivalent reformulation and confidence behavior under missing information are both relevant to evaluation and deployment. My interest depends on the paper being reframed more narrowly and evaluated more carefully. In the current version, the work feels incremental relative to the existing consistency and calibration literature, but the empirical pattern itself is still worth knowing.

**Claims And Evidence:**

No

**Claims Explanation:**

The core observation is plausible and mostly supported by the presented results: the paper's own tables and figures show sizable answer flip rates under perturbation while original and perturbed accuracy often remain close. That is a meaningful empirical point, and it supports the argument that accuracy alone is an incomplete reliability measure.

Several stronger claims are not supported. The manuscript states that the syntactic perturbations are motivated by socio-linguistic background variation, but the actual perturbation prompt only instructs Gemini to perform grammatical restructuring while preserving meaning. That prompt shows generic paraphrastic and syntactic variation only. It does not provide evidence about socio-linguistic background variation. The manuscript also claims that correct-to-incorrect transitions are the most critical failure mode for decision-support systems, provides no argument for why this is true. The analysis should treat correct-to-incorrect, incorrect-to-correct, and net paired accuracy change jointly rather than privileging one transition type by assertion.

The novelty claims are also overstated. The related-work discussion is sparse and misses direct prior work on consistency evaluation, semantic consistency, contrast-set robustness, prompt sensitivity, and elicited confidence calibration. That matters because the paper currently presents parts of the contribution as more novel than they are. A more accurate positioning would be: this is a simple multi-domain empirical study that combines several perturbation types and examines both answer flips and verbalized confidence in a critical-domain MCQ setting.

I also found the methodology underspecified. I could not find the exact evaluation prompt used to elicit answers, the answer extraction/parsing procedure, the confidence elicitation format and scale, or the handling of malformed outputs. That omission is serious in a prompt-sensitivity paper because part of the measured sensitivity can come from evaluation setting choices. The syntactic perturbation validation is also weaker than claimed: the manual audit already found a non-trivial fraction of bad or ambiguous rewrites, and the paper acknowledges that this may propagate to the final results. Finally, the experiments include only open-weight models, so the broad framing about "language models" as deployed decision makers is stronger than the evidence base.

**Requested Changes:**

Critical to my recommendation:

1. Reposition the paper against the direct consistency literature, and narrow the novelty claim accordingly. Relative to the current related-work section, which is quite sparse, the paper needs to engage with direct prior work on consistency evaluation and semantic consistency: BECEL (Jang et al., 2022), Consistency Analysis of ChatGPT (Jang & Lukasiewicz, 2023), Raj et al. on semantic consistency and reliability, Rabinovich et al. on predicting QA performance from semantic consistency, Johnson & Marasović on the joint interpretation of accuracy and consistency, and the recent Novikova et al. (2025) survey that maps the area explicitly. Given prior work, the claim that this paper "characterize[s] a previously under-measured reliability failure" is too strong.

2. Remove the socio-linguistic-background claim. The current perturbation pipeline demonstrates syntactic and paraphrastic variation only.

3. Do not present correct-to-incorrect transitions as the primary or most critical metric. Report incorrect-to-correct transitions and paired accuracy change, and discuss these jointly.

4. Add at least one strong proprietary model, or narrow the claims so they clearly apply only to the tested open-weight models. Gemini Flash 3.5 is fine and relatively affordable.

5. Fully specify the experimental protocol: exact prompts, confidence elicitation format and scale, answer extraction/parsing rules, handling of invalid outputs, model versions, API details, and perturbation-generation details.

6. Add a same-input repeated-run baseline to estimate residual instability that is unrelated to perturbation, and include uncertainty intervals or significance tests for the main comparisons. Maybe it turns out that zero temperature really means nothing changes on repeated runs, but that should be shown rather than assumed.

7. Tone down the deployment rhetoric around "decision makers" and "critical domains" unless the evaluation setup becomes closer to realistic decision-support workflows.

---

### Review · Reviewer_q3XU · 2026-06-26

**Summary Of Contributions:**

The authors evaluate models based on their susceptibility to perturbations in question presentation across seven domains in a multiple choice setting. The three perturbations tested are *invisible* (adding non-printing Unicode characters), *syntactic* (changing the syntax of the question without altering meaning), and *occlusion* (masking important tokens). The first two perturbations are meant to test whether models display decision invariance, whereas occlusion tests whether models display sensitivity to missing information. Ideally, models would show robustness to invisible and syntactic perturbations with no change in response confidence; conversely, occlusion should both decrease model accuracy and confidence.

Overall, they find that while overall accuracy remains relatively stable, perturbations can cause relatively high flip rates in settings where flipping should not be observed (with up to 18% flip rate for syntactic perturbations). They show that models’ verbalized confidence does not correspond with flip-rate, nor does confidence decrease when models flip to incorrect answers.

**Strengths**
- The paper is clearly written and easy to follow. Methods and metrics were clearly described.
- The analysis and visualization of flip rate vs. accuracy was interesting to observe, and supports the authors’ argument on the insufficiency of accuracy alone.

**Weaknesses**
- Many of the claims/results of this paper are already covered in existing literature—in particular, that LLMs are highly sensitive to prompts and that they have poor verbal confidence calibration—some of which are listed in the related work. This makes it unclear what the main contribution of the paper is. Another related work (“Do LLMs Exhibit Human-like Response Biases? A Case Study in Survey Design” by Tjuatja et al. 2024) also looks at the relationship between model confidence and susceptibility to question format and perturbations. Separately, there is a related line of work that is not discussed on the fragility of LLMs in multiple-choice settings, for example positional bias of answers (see “Large Language Models Are Not Robust Multiple Choice Selectors” by Zheng et al. 2024) and answer selection method (“When Benchmarks are Targets: Revealing the Sensitivity of Large Language Model Leaderboards” by Alzahrani et al. 2024), both of which use MMLU.
- I have concerns over two methodological points:
    - For the syntactic perturbation, only the question is provided to the LLM (Gemini-2.5-flash) that does the rewording (and similarly only the question before and after rewording is verified by humans); this can lead to an unintended semantic drift between the original intent of the question (which is made clear when paired with answers, especially if they are sentence completion). A clear example of this is in Table 1. When referencing the original question in MMLU where this came from, the answers to the original question are rates of photon detection, not absolute quantities (which is what the syntactic perturbation rewords the question to). This means that for some 	questions, semantic equivalence is not preserved.
    - It is unclear why the authors decided to measure confidence through models’ verbalizations (which are known to be often poorly calibrated) as opposed to measuring it through something like entropy over the label space (as was done in Tjuatja et al. 2024).

**Audience:**

Yes

**Audience Explanation:**

Given that the authors will be able to clarify their findings and how they are separate from existing work, the use of LLMs in various decision-making settings is relevant to many practitioners, including (but not limited to) as decision support/tools for augmented decision making, and in settings like LLM-as-a-judge/verifiers. Understanding the bounds of their robustness is important across these use cases.

**Claims And Evidence:**

No

**Claims Explanation:**

Some, but not all, claims of the paper are supported. As pointed out above, because the syntactic perturbation may not actually be preserving semantics in some (unknown quantity) of questions in the dataset, it is hard to tell how much of the effects observed in the syntactic setting are due to semantic change versus surface-level changes. Additionally, the claim of confidence may change based on the metric used to measure model response confidence.

**Requested Changes:**

Following up on the weaknesses and concerns listed above (listed in order of importance):
- Clarify the main contribution of the paper, in particular how the findings expand upon or differ from existing related work
- Validate the semantic invariance of the syntactic perturbations by making sure that the answer continuations are consistent with the reworded question
- Either justify the use of verbalized confidence measures, or add an analysis of confidence that is logit/label distribution based

---

### Review · Reviewer_smud · 2026-06-30

**Summary Of Contributions:**

The authors focus primarily on evaluation of whether model predictions remain stable under perturbations of the prompt, where the property of decision invariance is critical for realistic assistive decision-making settings.
The authors consider a QA task for decision-support, where the questions are perturbed.
the authors experiment with three perturbations: (1) randomly adding non-printing unicode characters, (2) prompting a model re-write questions while preserving meaning, and (3) omitting keywords. The authors find that models exhibit instability across all domains in their experiments.

Strengths:
- Sound experimental setup over multiple model sizes and architectures with respect to measurement of prompt sensitivity (flip rate and correct-to-incorrect rate).
- Focused on a critical problem space of evaluation for realistic, decision-support settings. Authors put forth a clear and concrete value to measure: decision-invariance with obvious importance for decision-support.
- Clear and well-written.

Weaknesses
- The major weaknesses of this work are that (1) it is not reflective of a decision-support setting and that (2) the measurement of confidence is not meaningful (see response to following question for more details).
- The work would be strengthened with more discussion on the users of decision-support systems. For example, the intuition for the syntactic perturbation is that "people present/structure information differently based on their socio-linguistic background." Similarly, connections between grammatical formulations and people's background as discussed in section 4 are not grounded in evidence. It is not evident to me how the perturbations are tailored to use of LMs for decision-support. For example, I can imagine that clinicians that are trained similarly may share the same writing style, and this training may be standardized for the field. Similarly, for the occlusion perturbation, it is unclear to me that omitting key phrases is representative of realistic patterns. The paper would be much stronger if the perturbations were grounded in direct observation of (/prior work on) usage of decision-support systems.
- It is unclear that keywords extracted with RAKE are necessarily informative for answering the questions. Verifying this manually for a subset would strengthen the validity of the occlusion perturbation. Relatedly, it may be worth exploring if there are cases where masking a word reduces ambiguity if there is some extraneous information presented in the question.

**Additional Comments:**

Questions:
- What is a question stem? The original question.
- Why do you think the larger models exhibit a high flip-rate? This seems to contradict ProSA from Zhuo et al. 2024.

**Audience:**

Yes

**Audience Explanation:**

The authors experiments with invariance-type perturbations highlight very high sensitivity to superficial input variation. And realistic variation in prompt likely exhibit a wider range, so this paper represents a lower bound. This certainly has implications for high-stakes decision-support systems that incorporate models.

**Claims And Evidence:**

No

**Claims Explanation:**

Unclear that this paper is reflective of a decision-support setting, which is the expressed scope of this work.
- For one, it is unclear that the perturbations are actually well-aligned with how LMs for decision support are actually used.
- Second, while the MedQA dataset is appropriate, the MMLU dataset may not reflect realistic questions for decision-support questions.

Measurement of model confidence is not meaningful.
- The confidence scores reported in the paper are elicited by prompting a model. This is concerning, as the conclusions about occlusion perturbations producing instability depend on this measurement.
- Since the authors are using multiple choice questions, a reasonable (although still limited) measurement of confidence could be a function of the log probabilities of the choices (where each choice can be represented as a token).

**Requested Changes:**

In order to support claims about occlusion perturbation, it would be necessary to estimate confidence scores.

Alternatively, I think a strong paper could also be focused solely on invariance-type perturbations, where the perturbations are designed to reflect prior work on realistic, natural language variation. Ideally, these perturbations would be domain-specific.

---

### Author Response · Authors · 2026-07-03
**General response to reviewers (1/2)**

We thank the reviewers for their constructive feedback. We identified common concerns across the reviews and grouped them below, along with our responses outlining how we will address each.

**1. Novelty is not clearly positioned against the existing consistency-evaluation literature (q3XU, 1otq)**

Response: We will rewrite Section 2 to explicitly position this work against the following: BECEL (Jang et al., 2022), Consistency Analysis of ChatGPT (Jang & Lukasiewicz, 2023), Raj et al. on semantic consistency and reliability, Rabinovich et al. on predicting QA performance from semantic consistency, Johnson & Marasović on joint interpretation of accuracy and consistency, the Novikova et al. (2025) survey, Tjuatja et al. (2024) on response biases and question-format sensitivity, Zheng et al. (2024) on positional bias in multiple-choice selection, and Alzahrani et al. (2024) on answer-extraction sensitivity in MMLU leaderboards.
We will clarify this distinction and position our work as a joint evaluation of three qualitatively different perturbation classes ranging from those with zero semantic shift to those that delete core information against self-reported confidence, in order to assess whether self-reported confidence is a reliable proxy for consistency. To ensure a more defensible positioning of our contributions, we will remove the phrase "characterize a previously under-measured reliability failure" from both the abstract and introduction, replacing it with a more focused claim centered on the joint analysis of stability and calibration across varying levels of perturbation severity.

**2. The decision-support framing and the socio-linguistic background claim are not grounded in evidence (smud, 1otq)**

Response: We will remove the socio-linguistic background claim from Section 4 and reframe the motivation for the syntactic perturbation purely in terms of established prompt-sensitivity literature (Sclar et al., 2024, already cited), rather than a claim about who asks questions and why. We will also pass over the abstract, introduction, and discussion to soften the terms "decision makers," "critical domains," and similar deployment language, so that they match what the paper actually shows: that MCQA benchmark accuracy does not capture prediction stability under controlled input perturbations.

**3. Syntactic perturbations may not preserve semantic equivalence, because generation and verification only used the question stem (q3XU)**

Response: We revisited the specific item underlying Table 1; the question and its options are reproduced below.

Stem: "The quantum efficiency of a photon detector is 0.1. If 100 photons are sent into the detector, one after the other, the detector will detect photons ___"

Options:

A: an average of 10 times, with an rms deviation of about 4

B: an average of 10 times, with an rms deviation of about 3

C: an average of 10 times, with an rms deviation of about 1

D: an average of 10 times, with an rms deviation of about 0.1

These are mean-and-spread descriptions of a detected-photon count, not rates. The original stem is a fill-in-the-blank construction ("the detector will detect photons ___"), and the syntactic rewrite turns this into an explicit question ("How many photons will it detect?"). Both are asking for the same thing: the distribution of the number of photons detected. We do not see a change in the correct answer for this item, and we could not find a basis for describing the original answer choices as rates rather than as a count distribution.
Our generation prompt for syntactic perturbation already instructs the model to "not change any technical terms, named entities, or numerical values" and to "assume the answer options remain exactly the same and that the correct option must remain correct." We acknowledge this as a genuine limitation of our syntactic-perturbation pipeline rather than treating it as resolved, as noted in our Limitations section. We mitigate this risk with three safeguards: generation instructions, semantic-similarity filtering, and human evaluation.
The suggestion of manual stem-pair verification is valuable, but it falls outside the scope of the current project; we will note it as future work in our Limitations section.

---

> ### Author Response · Authors · 2026-07-03
> **General response to reviewers (2/2)**
>
> **4. RAKE-selected occlusion tokens are not verified to be informative for answering the question (smud)**
>
> Response: Reviewer smud notes that we never verify that RAKE keyword extraction actually selects tokens that matter for answering the question, and asks that this be checked manually on a subset; they also raise the related possibility that masking a word could sometimes reduce ambiguity rather than increase it, if the question stem contains extraneous information.
>
> In response, we will manually annotate 100 randomly sampled masked tokens, mirroring the scale of our syntactic verification. We will categorize each item into whether (1) the token is essential for a correct response, (2) the token is redundant or contextually inferable, or (3) occlusion actually removes extraneous noise rather than vital information. We will report the resulting distribution in the appendix alongside our other manual-annotation results, and discuss in the main text how it should be interpreted in relation to the occlusion flip-rate results.
>
>
> **5. Verbalized confidence is a weak calibration signal and was not justified against alternatives (smud, q3XU)**
>
> Response: For Llama-3.1-8B and Gemma-2-9B, which we host locally and for which log-probabilities are available at no additional cost, we will compute an entropy-based confidence measure directly from the output-token log-probabilities over the answer options, and report it alongside verbalized confidence.
> For Llama-Scout-17B and Kimi-K2-1T, served via the Groq API, we will not run the equivalent log-probability-based experiment, since it would require re-querying our full evaluation set (4,098 questions × perturbations × repeats) through a paid API purely to extract per-token probabilities, at a cost disproportionate to the marginal evidence gained.
> We would also like to note Tian et al. (2023, "Just Ask for Calibration," EMNLP), who show that for RLHF models — the same model class as all four models we evaluate — verbalized confidence is typically better calibrated than the model's conditional token probabilities on open-domain QA, reducing expected calibration error by a relative 50%.
>
> **6. Correct-to-incorrect transitions are overweighted relative to incorrect-to-correct and paired accuracy (1otq)**
>
> Response: We will add an I → C column to Table 2 to complement the existing C → I data — this is a re-analysis of existing predictions rather than a new experiment — and display the paired accuracy delta (the difference between perturbed and original performance on the identical item set) immediately adjacent to both columns. Within the Results and Discussion sections, we will jointly evaluate C → I and I → C transitions for every perturbation class. We will also update the text to avoid implying that C → I is the sole failure mode; instead, we will explicitly state that, while the flip rate remains high, the net accuracy shift is minimal because C → I and I → C transitions occur with similar magnitudes for perturbations that preserve information.
>
> **7. Model coverage is limited to open-weight models; claims may not generalize to proprietary models (1otq)**
>
> Response: We will add Gemini-2.5-Flash as a fifth evaluated model, run through the identical perturbation and evaluation pipeline used for the other four.
>
> **8. The experimental protocol is underspecified (1otq)**
>
> Response: We will add an appendix documenting the exact prompt templates used for answer and confidence elicitation, the answer-parsing logic, and how malformed outputs are handled.
>
> **9. No baseline exists to separate perturbation-driven instability from residual model/serving noise (1otq)**
>
> Response: We will run a repeated-run baseline on a subsample of approximately 150–200 original (unperturbed) questions, executing each 3–5 times per model under the same serving configuration used for the main experiments, and report our findings.

---

> > ### Comment · Reviewer_1otq · 2026-07-03
> >
> > Thank you for the detailed response. The planned revisions address many of my concerns. Since the submission PDF is unchanged, my assessment of these items is conditional on the changes appearing in the revised manuscript.
> >
> > I have a few concerns remaining:
> >
> > Full-item validity of the syntactic perturbations (your point 3; my main remaining concern). Your defense of the Table 1 photon item may be correct, but the issue is systematic rather than about one example. The relevant unit of semantic equivalence in MCQA is not the stem alone; it is the stem together with the fixed answer options and the induced correct answer. A rewrite can preserve the stem's apparent meaning while changing how the options attach to it, or which option is best. None of the three safeguards you list tests this property: the generation-prompt constraint is unverified model compliance, the similarity filter compares stems only, and the 100-item human audit compared stems only. Since the paper's central invariance claim rests on these rewrites being answer-preserving, I do not agree with "outside the scope of the current project"; especially given that you already ran a 100-item stem audit and are planning a 100-item RAKE audit. The needed audit is the same size: sample ~100 perturbed items, show annotators the original stem with its options and the perturbed stem with the same options, ask whether the correct answer is preserved, and report good/bad/ambiguous rates under this full-item criterion. If the bad-or-ambiguous rate again lands near 10%, that is on the same order as the syntactic flip rates (12–18%), and the syntactic results would need substantially more cautious interpretation. This audit, or a substantially narrower syntactic-invariance claim, remains critical to my recommendation.
> >
> > Confidence claims need narrowing (point 5). Entropy-based confidence for the two locally hosted models is a good addition, and the cost argument for the API-served models is reasonable. But the revision will then have verbalized confidence for five models and log-probability-based uncertainty for only two, and the claims must track that asymmetry. Tian et al. (2023) does not settle this either: verbalized confidence sometimes outperforming conditional token probabilities on open-domain QA does not establish it as a validated calibration signal under these perturbations. I think the right claim is narrower: in this setup, models often maintain high verbalized confidence even when their answers change or become wrong.
> >
> > Gemini-2.5-Flash (point 7): useful, but it also generated the syntactic perturbations. Disclose this when reporting its results and note the possible interaction between the perturbation generator and an evaluated model. OR (better) use another model entirely like Haiku.
> >
> > Uncertainty: this part of my original change 6 was not addressed. Add bootstrap confidence intervals (resampled at the question level, since perturbed instances cluster within items) for the main flip rates, transition rates, accuracy deltas, and the domain comparisons (e.g., law/ethics vs. biology).

---

> ### Author Response · Authors · 2026-07-04
>
> We thank the reviewer for the detailed follow-up and for engaging closely with the safeguards we described. Below we have addressed your concerns.
>
> >Confidence claims need narrowing (point 5).
>
> We agree, and we will adopt the narrower claim based on our findings from entropy-based confidence estimates.
>
> >Gemini-2.5-Flash (point 7):
>
> We agree that it should not be Gemini-2.5-Flash. The bigger model would be either from the Claude family or the GPT family.
>
> >Uncertainty
>
> We agree, and our answer previously was incomplete. Below is our complete plan:
>
> We will replace the current instance-level treatment with a cluster (block) bootstrap: for at least 2,000 replicates, we resample question indices with replacement, include all perturbation instances for each resampled question, and recompute the statistic on the resulting pseudo-sample. We will report 95% percentile bootstrap intervals, computed this way, for:
>
> 1. the main flip rates for each perturbation type and model,
> 2. the correct-to-incorrect and incorrect-to-correct transition rates (both re-sliced from existing predictions, as in our response to change 6),
> 3. the paired accuracy deltas (perturbed minus original accuracy on the same item set), and
> 4. the domain comparisons referenced by the reviewer (e.g., law/ethics vs. biology), where we will bootstrap the difference in each statistic between domain item pools and report whether the resulting interval excludes zero.
>
> >Full-item validity of the syntactic perturbations (your point 3; my main remaining concern)
>
> We understand this concern and share the reviewer's perspective. However, we do not believe manual annotation of stem+options would be helpful, as certain aspects of the experiment were designed to make this addition infeasible. To elaborate, for syntactic rewrites, only the stem/question text was provided to the language model, and we do this based on the existing standard of writing MCQ questions, as discussed in existing literature (Haladyna, Downing, and Rodriguez's (2002) A Review of Multiple-Choice Item-Writing Guidelines for Classroom Assessment; NBME Item-Writing Guide; Case & Swanson). Both guidelines state that the question text should stand independently of the options (answers or distractors); hence, our experiments operate only at the question level.
>
> However, we are open to making the following changes:
>
> 1. We will explicitly change our narrative and state that the syntactic rewrites are verified as meaning-preserving with language that describes them as phrasing perturbations whose surface-level (stem-only) similarity to the original was checked, and whose answer-preservation was not independently verified against the fixed options.
>
> 2. We will state explicitly, next to the syntactic flip-rate results (currently 12–18% across models), that this rate should be read as an upper bound on genuine model instability under phrasing change, since an unknown fraction may instead reflect undetected answer-attachment drift introduced during rewriting.
>
> 3. We will add more details about the assumptions underlying the syntactic perturbation experiment, based on the aforementioned literature in measurement science.
>
> A more cautious, narrower claim about syntactic invariance, grounded in established item-writing guidelines, should address the primary concern.

---

> > ### Comment · Reviewer_1otq · 2026-07-06
> >
> > My main remaining concern is still the full-item validity of the syntactic perturbations. I understand the appeal to item-writing guidelines and the cover-the-options rule. Those guidelines do support the idea that a well-written MCQ stem should usually be answerable without looking at the options. But they do not establish that all items in these benchmark subsets satisfy that property, and they do not establish that an LLM-generated rewrite of the stem preserves the answer relation to the original fixed options.
> >
> > So I do not agree that full-item annotation would be unhelpful or infeasible. The fact that the generator only saw the stem is precisely why a full-item audit would be useful. The audit would not require regenerating perturbations or rerunning models. It would only require sampling original/rewritten item pairs and asking whether the original keyed answer remains correct under the rewritten stem and fixed options.
> >
> > That said, I appreciate the proposed narrowing. If the authors explicitly describe the syntactic perturbations as stem-only phrasing perturbations, state that answer preservation against the fixed options was not independently verified, and present the 12–18% syntactic flip rate as an upper bound on genuine model instability under phrasing change, then my concern is substantially reduced. This language should appear not only in the limitations section, but also next to the main syntactic results and in any abstract/introduction claim that summarizes those results.